# Cardiac Tumors Causing Sudden Cardiac Death: A State-of-the-Art Review in Pathology

**DOI:** 10.3390/cancers17040669

**Published:** 2025-02-17

**Authors:** Cecilia Salzillo, Stefano Lucà, Andrea Ronchi, Renato Franco, Giulia Iacobellis, Alessia Leggio, Andrea Marzullo

**Affiliations:** 1Department of Experimental Medicine, PhD Course in Public Health, University of Campania “Luigi Vanvitelli”, 80138 Naples, Italy; stefano.luca@unicampania.it; 2Pathology Unit, Department of Precision and Regenerative Medicine and Ionian Area, University of Bari “Aldo Moro”, 70121 Bari, Italy; 3Pathology Unit, Department of Mental and Physical Health and Preventive Medicine, University of Campania “Luigi Vanvitelli”, 80138 Naples, Italy; andrea.ronchi@unicampania.it (A.R.); renato.franco@unicampania.it (R.F.); 4Radiology Unit, Department of Clinical and Experimental Medicine, University of Foggia, 71122 Foggia, Italy; giulia.iacobellis@unifg.it; 5Legal Medicine Unit, Department of Interdisciplinary Medicine, University of Bari “Aldo Moro”, 70121 Bari, Italy; a.leggio@eufor.eu

**Keywords:** cardiac tumors, benign tumors, malignant tumors, sudden cardiac death

## Abstract

Cardiac tumors are a rare and diverse category of neoplasms that represent a significant challenge for diagnosis and treatment. This review aims to provide an updated overview of the state of the art of cardiac tumors, with a focus on histopathological and molecular features. Furthermore, it underlines the importance of a better understanding of the biological and behavioral characteristics of these tumors, both benign and malignant, to improve early diagnosis and personalize therapeutic strategies. The review emphasizes the importance of a multidisciplinary team to improve the outlook for patients with these rare but potentially lethal tumors, to prevent fatal events such as sudden cardiac death.

## 1. Introduction

Cardiac tumors (CTs) are tumors that affect all cardiac structures such as the myocardium, valves and cardiac chambers [1]. The first CT was described by pathologist Realdus Columbus and the first diagnosis was made in 1934 [2].

CTs are divided into primary cardiac tumors (PCTs) and secondary cardiac tumors (SCTs) or metastases.

PCTs are rare with a prevalence of 0.02% at autopsy, of which benign cardiac tumors account for 75% and malignant heart tumors the remaining 25% [3]. SCTs are more frequent but are often diagnosed with autopsy as they are asymptomatic with a frequency of 0.4% and in patients with diagnosed cancer there is cardiac involvement of 20% [4].

Symptoms vary depending on the nature, location, and size of the tumor; however, they can also be asymptomatic [5].

Tumors that interfere with blood flow manifest obstructive symptoms such as dyspnea, syncope and pulmonary edema. When fragments of the tumor break off, they manifest with embolisms and when they compress or infiltrate the conduction pathways with arrhythmias. Rapid growth and invasion of surrounding tissues of malignant cardiac tumors can cause pericarditis and cardiac tamponade. Furthermore, non-specific symptoms such as fever, asthenia, and weight loss may be present, which often trigger a systemic inflammatory response [6,7].

Diagnosis is made through clinical evaluation, instrumental tests, and histological examination.

Transthoracic echocardiogram (TTE) is the first step to identify abnormal masses within the heart chambers, while transesophageal echocardiogram (TEE) allows a more detailed view of areas that are difficult to explore. Magnetic resonance imaging (MRI) is essential for characterizing the tumor, distinguishing between benign and malignant formations and evaluating the extension to the surrounding tissues. Computed tomography (CT), on the other hand, is often used to investigate the presence of metastases or to plan surgery. Left heart catheterization can be helpful in determining blood supply and structures adjacent to tumors, tumor invasion into epicardial vessels, and, with ventriculography, tumor extension into the ventricular chamber can be determined [4,7,8].

Definitive diagnosis requires histological examination, often performed after surgical resection. Intracardiac biopsy is a risky procedure and is performed in specialized centers with expert staff, especially for tumors on the right side of the heart which have fewer complications than the left side, where the risk of perforation and embolization increases. Imaging-guided techniques improve biopsy success and sample quality. Pericardial fluid cytology can also identify the nature of the tumor, while surgical resection with histological and fluoroscopic analysis allows the classification of the neoplasm [4,9].

Treatment depends on the type of tumor. Benign cardiac tumors are usually surgically removed, with good results and a favorable prognosis. For malignant CTs, however, the treatment is more complex with a combination of surgery, chemotherapy, and radiotherapy, but the prognosis often remains unfavorable with limited survival [6,10].

In this state-of-the-art review, we focus on the histological characteristics of benign and malignant CTs and the association with SCD.

## 2. Classification

The fifth edition of the WHO Classification of Thoracic Tumors [11] updates the categorization of CTs, subdividing them into benign, malignant, and hematolymphoid neoplasms (Table 1) and introduces important updates on CTs [12].

One of the main novelties is the separation of intimal sarcoma and undifferentiated pleomorphic sarcoma into distinct entities, which reflects a better understanding of their biology and their site of origin. Some rare tumors, such as rhabdomyosarcoma and osteosarcoma, have been moved to mesenchymal tumors.

Papillary fibroelastomas are considered neoplastic and represent the most frequently removed cardiac tumor with a frequency almost double that of cardiac myxoma.

Histiocytoid cardiomyopathy was reclassified as hamartoma of the conduction system, and cardiac mesenchymal hamartoma and lipomatous hamartoma of the atrioventricular valves were added to the classification for distinctive clinical and pathologic presentations.

Additionally, a section on genetic tumor syndromes has been introduced.

## 3. Benign Tumors

### 3.1. Papillary Fibroelastoma

Papillary fibroelastoma (PFE) is the most frequent benign cardiac neoplasm, with a risk of embolization and consequently myocardial infarction, stroke, fibrillation, and SCD [13].

The incidence of PFE is underestimated as they are usually diagnosed with postmortems at autopsy [14]. The most frequent site of PFE is the aortic valve in 35–63% of cases, the mitral valve in 9–55%, the tricuspid valve in 6–15%, and the pulmonary valve in 0.5–8% [15].

PFEs are frequent in men over the age of 40 with the majority diagnosed by age 60 but are also described in infants and young children especially with congenital cardiac anomalies [16].

The etiopathogenesis of PFE is not completely known, but it is believed to be multifactorial in nature, such as genetic, mechanical, and biological factors.

One hypothesis holds that PFE results from acquired lesions, which originate from microthrombi that adhere to minor endothelial damage on the valve surfaces and, over time, evolve into growths to form the neoformation [14].

Another theory suggests a congenital or malformative origin, but the recent identification of oncogenic mutations in KRAS has led to the hypothesis that PFE may have a neoplastic origin, although the clinical behavior is benign [12,17].

Clinically, most PFE is asymptomatic and is diagnosed incidentally during cardiac imaging tests, such as echocardiography or computed tomography performed for other pathological indications [14]. In a minority of cases, PFE can manifest itself with embolic events, of which the most common are transient ischemic attacks and strokes, followed by myocardial infarction, syncope, pulmonary, or peripheral embolisms and sudden cardiac death [3,14,18,19].

Histological examination is essential for the definitive diagnosis of PFE, but the initial evaluation is generally carried out with echocardiography with diagnostic accuracy above 85% [16].

With an echocardiography, PFE is characterized by a small <1.5 cm, round, echo-dense, pedunculated mass with high independent motion, with a glistening appearance related to thread-like projections [16].

Macroscopically, the PFE appears as a small round or oval mass, generally pedunculated, with dimensions ranging from a few millimeters to approximately 2 cm in diameter. The surface of the PFE is characterized by numerous thin and flexible papillary projections, lined by endocardium and appearing avascular, giving an appearance like that of a sea anemone [12,14].

Histologically, PFE is characterized by fronds covered by a single layer of endothelial cells that cover a central core of avascular fibroelastic connective tissue. The core is composed of an extracellular matrix rich in collagen, elastin, and proteoglycans. Endothelial cells do not show signs of atypia or obvious neoplastic proliferation (Figure 1) [12,14].

In some cases, it is possible to observe small deposits of fibrin on the surface, especially in the presence of microtrauma or local hemodynamic turbulence, which can increase the risk of embolic events [12,14].

The risk of embolic complications of PFE justifies a therapeutic approach aimed at surgical excision [18,19], making it the most frequently surgically removed cardiac tumor, almost twice as often as cardiac myxoma.

### 3.2. Myxoma (NOS)

Myxoma is a benign cardiac tumor that can cause serious morbidity and mortality if not treated promptly due to the risk of intracardiac obstruction and systemic emboli resulting in SCD [20,21].

It accounts for 50–85% of PCTs [22], with an incidence of 0.03% in the general population [21]. It occurs between the fourth and seventh decade of life [23], mainly affecting women and with 75% of cases involving the left atrium [24].

Myxoma is asymptomatic in 20% of cases [23], but frequently manifests itself with obstructive symptoms such as dyspnea and syncope, which simulate mitral stenosis when they interfere with the blood flow between the atrium and ventricle, and the mobility of the mass can cause systemic embolisms, with fragments that reach the cerebral, peripheral, or pulmonary circulation in tumors of the right atrium, leading to stroke, ischemia, or pulmonary embolism [25].

Furthermore, it can manifest itself with constitutional symptoms such as fever, asthenia, weight loss, caused by the production of interleukin-6, which contributes to a systemic inflammatory framework [26].

The diagnosis of myxoma is based on instrumental investigations and confirmation with histological examination.

Upon echocardiography it appears as a spherical mass attached to the endocardial surface with areas of echogenic calcification, while on MRI and CT often appear heterogeneous and with different shapes [3].

Macroscopically, myxoma appears as a lobulated or villiform, friable mass, with areas of hemorrhage and necrosis, and of variable size (Figure 2A,B). Histologically, it is characterized by isolated or grouped cellularity, immersed in a myxoid matrix typically rich in proteoglycans, collagen, and elastin (Figure 2C). In most cases it is calretinin, vimentin, S100, nonspecific enolase, factor VIII, CD31, and CD34 (Figure 2D) positive on immunohistochemistry [3,12,25].

The therapy is exclusively surgical and consists of the complete removal of the tumor, including its implant base to prevent recurrence.

The prognosis is generally excellent in sporadic cases, but in less than 10% these are familial myxomas with a risk of recurrence associated with Carney complex syndrome (CNS) which is a rare autosomal dominant disease caused by germline mutations in the PRKAR1A gene and characterized by myxomas, skin pigmentation, and endocrine hyperactivity [27,28,29].

### 3.3. Fibroma (NOS)

Fibroma is a rare benign cardiac tumor that can cause intracavitary obstruction, inflow and outflow tract obstruction, coronary artery impairment, thromboembolic events, conduction defects, and consequently SCD, especially in pediatric patients [30,31,32,33].

Cardiac fibroma is very rare, with an incidence of approximately one case in 280,000 live births. It is the second most common in pediatric age, preceded only by rhabdomyoma, and represents 25–30% of benign cardiac tumors, and is the second most common tumor in fetuses. It affects both sexes equally [30,34].

The etiopathogenesis of cardiac fibroma is unclear. In a minority of cases, it is associated with Gorlin–Goltz syndrome, an autosomal dominant disease caused by mutations in the tumor suppressor genes PTCH1 and SUFU, which regulate the Hedgehog signaling pathway [35,36]. However, most cases of cardiac fibroma have no known genetic associations, and it is hypothesized that the tumor grows in proportion to the physiological development of the heart, reaching considerable size in older children.

Clinical manifestations vary depending on the tumor’s location, size, and interference with cardiac function. In some cases, the fibroid is asymptomatic and discovered incidentally during instrumental investigations. If symptomatic, it can manifest with arrhythmias such as ventricular tachycardia or ventricular fibrillation, intracardiac obstruction, conduction blocks, and SCD [30,31,32,33].

The diagnosis of cardiac fibroma is mainly based on imaging techniques and histological examination.

On echocardiography, it is the first-choice examination, and the fibroma appears as a well-circumscribed, hyperechogenic, and sometimes calcified mass. On MRI, it appears hypointense on T2 sequences, with little enhancement after the administration of contrast medium. CT identifies calcifications. The ECG can highlight alterations in the heart rhythm [30].

Macroscopically, the fibroma appears as a grayish-white, solid, non-encapsulated mass, often with visible calcifications. On histology, it is characterized by uniform fibroblasts immersed in an abundant collagen matrix, and in children there are extensive calcifications that increase with age [37,38,39].

Treatment of cardiac fibroma depends on the symptoms and risk of complications. Complete surgical resection is the therapy of choice and is frequently curative. If total removal is not possible, even partial resection may improve symptoms. Furthermore, in patients at a high risk of malignant arrhythmia an implantable defibrillator may be necessary, and cardiac transplantation in inoperable cases or cases with severe functional impairment [30].

### 3.4. Rhabdomyoma (NOS)

Cardiac rhabdomyoma (CR) is a rare benign mesenchymal tumor that arises from striated muscle, is a type of hamartoma, and can cause left ventricular outflow tract obstruction and malignant ventricular tachycardia, resulting in SCD [40,41,42,43].

CR represents the most frequent pediatric cardiac tumor—approximately 45% of cases—with a uniform distribution between the sexes and manifests mainly within the first year of life [40].

CR mainly affects the ventricular myocardium, the atria, the cavo-atrial junction, and the epicardial surface [44,45].

CR is associated in 80–90% with tuberous sclerosis, an autosomal dominant disorder characterized by multiple benign hamartomas in various organs, but no other causes are known [46,47].

Clinically it is heterogeneous, and in most cases, it is asymptomatic and diagnosed incidentally via prenatal ultrasound. When symptomatic, it manifests congestive heart failure, arrhythmia such as bradycardia, ventricular or atrial tachycardia, obstruction of the ventricular inflow/outflow tract, and SCD. In newborns, heart blockage or hydrops fetalis may occur [40,41,42,43].

The diagnosis of CR is made by imaging and histology.

Echocardiography is the first-line examination and shows multiple, homogeneous, hyperechogenic masses in the ventricular myocardium. In complex cases, MRI can improve the characterization of the masses and assess ventricular function. ECG is useful for detecting arrhythmia or conduction abnormalities [48].

On macroscopic examination, CR appears as multiple well-circumscribed nodules, which develop primarily in the myocardium (Figure 3). Histologically, it is characterized by spider cells, with cytoplasmic vacuolation and anomalies of cellular architecture, characteristic of their hamartomatous nature [48].

Most CR regresses spontaneously so treatment is conservative. In patients with tuberous sclerosis, emerging drugs are mTOR inhibitors [49]. Surgery is only indicated in the presence of severe symptoms or significant obstruction, while heart transplant is extremely rare.

### 3.5. Adult Cellular Rhabdomyoma

Adult cellular rhabdomyoma (ACR) is a rare benign neoplasm that may manifest with electrophysiological disturbances, such as supraventricular tachycardia or intermittent ventricular tachycardia and SCD [50,51].

ACR affects individuals older than 20 years of age, predominantly the atrium and is not associated with genetic syndromes unlike the pediatric form [12].

Macroscopically, it appears as homogeneous, beige, circumscribed masses that may invade adjacent cardiac chambers [52].

Histologically, it is composed of striated muscle cells reminiscent of extracardiac rhabdomyomas and is more cellular, with growth in sheets of ovoid to spindle-shaped striated myocytes, immersed in a rich vascular background, and lacks the characteristic vacuolated “spider cells” typical of the infantile form [52].

Immunohistochemistry is positive for desmin and myogenin, while cellular proliferation is low with a Ki-67 index between 1% and 20% [52].

### 3.6. Lipoma (NOS)

Lipomas are rare benign cardiac tumors that can develop in any layer of the heart, with a predilection for the endocardium of the right atrium, the left ventricle, and the pericardium [3], rarely associated with SCD, probably due to arrhythmogenic or obstructive complications [53,54].

The etiopathogenesis is not entirely clear, but risk factors include a high body mass index, advanced age, and female sex [3].

The incidence of cardiac lipoma is 2.4% of all primary heart tumors [54].

Cardiac lipomas are usually asymptomatic, but whether they are symptomatic depends on the size and location of the mass, with dyspnea, chest pain, arrhythmia, and obstruction of blood flow [55].

Diagnosis is based on symptoms and imaging studies, with echocardiography and CT or MRI [7,56].

Macroscopically, it presents itself as a well-circumscribed and encapsulated mass; microscopically it is characterized by mature adipocytes free of atypia (Figure 4) [12].

In symptomatic and large lipomas, the treatment is surgical resection, whereas for asymptomatic or small lipomas a conservative approach with regular monitoring is recommended [57,58].

### 3.7. Lipomatous Hypertrophy of the Atrial Septum

Lipomatous hypertrophy of the atrial septum (LHAS) is a rare benign cardiac tumor [59] and may be associated with SCD, due to severe arrhythmia, critical obstruction of blood flow, or embolic complications [60,61].

Etiopathogenesis is not fully understood, but it is thought to be caused by a hyperplastic process of the septal adipose tissue and may present genetic alterations like those of lipomas.

LHAS has a prevalence between 1 and 8%, and the incidence increases with age, body mass, and chronic corticosteroid therapy, and is often discovered incidentally being asymptomatic during cardiac imaging studies, surgery, or autopsies [62].

Symptoms vary according to the size and location of the fat accumulation. LHAS is usually asymptomatic, but it may cause symptoms such as obstruction of blood flow of the vena cava or ventricular outflow tract, arrhythmia, and heart failure [63].

Diagnosis is mainly based on cardiac imaging. Echocardiography is the first diagnostic approach, but MRI allows better definition of the margins and extent of the lesion in the interatrial septum and surrounding tissues [64].

Macroscopically, LHAS appears as a non-encapsulated mass, predominantly in the interatrial septum, which may extend to other cardiac structures, causing obvious anatomical deformities [12,64].

Microscopically, LHAS consists of a mixture of adipose tissue including interspersed by fetal fat cells, fibrous tissue, and hypertrophied myocytes, without evidence of atypia or malignancy (Figure 5). LHAS is differentiated from cardiac lipoma by the absence of encapsulated adipose cells and the presence of fetal fat cells [12,65].

Treatment depends on the symptoms. If asymptomatic, clinical and radiological monitoring is sufficient, whereas if symptomatic, surgical resection is indicated [65].

### 3.8. Lipomatous Hamartoma of Atrioventricular Valve

Lipomatous hamartoma of the atrioventricular valve (LHAV) is a rare benign cardiac tumor described in the fifth edition and probably congenital hamartomatous expansions of the mitral and tricuspid valves [12].

It is usually asymptomatic, but may manifest with valvular regurgitation, embolic symptoms or hemodynamic alterations in relation to the position of the mass [66,67].

Macroscopically, LHAV is characterized by thickened and prolapsing atrioventricular leaflets with nodular protuberances [12].

Histologically, LHAV is characterized by a hamartomatous expansion composed of disorganized mature adipose tissue, fibroconnective tissue, and thin-walled vasculature. It is distinguished from cardiac lipoma by the absence of a capsule and therefore by an infiltrative or expansive appearance under the microscope [66,68].

The therapy consists of surgical excision, and if possible, with valve preservation [67].

### 3.9. Hamartoma of Mature Cardiac Myocytes

Hamartoma of mature cardiac myocytes (HMCM) is a rare benign primary cardiac tumor, characterized by disorganized growth of mature cardiac myocytes [69].

The pathogenesis is unclear, but they may result from congenital embryonic dysplasia [70].

HMCM usually arises from the free wall of the left ventricle, but can also be localized on the interventricular septum, on the wall of the right ventricle or in the right atrium [69] and involves the pulmonary infundibulum [71]. It affects ages between 6 months and 76 years [71].

Clinically, HMCM can be asymptomatic and discovered incidentally, but in some cases, it manifests itself with dyspnea, chest pain, incessant ventricular tachyarrhythmias, and SCD [69].

Diagnosis is made by echocardiography and cardiac magnetic resonance imaging, and confirmation is histological [72].

Macroscopically, HMCM appears as a pale-gray, ill-defined myocardial mass; histologically, it is characterized by hypertrophic myocytes in disorganized arrangements arranged in a herringbone pattern, random, pinwheels, or spiral and interstitial fibrosis [12,73].

Surgical resection is recommended to avoid complications such as intractable arrhythmias or sudden death [73].

### 3.10. Mesenchymal Cardiac Hamartoma

Cardiac mesenchymal hamartoma (CMH) is introduced in the fifth edition and is a rare benign neoplasm composed of cardiomyocytes with a disorganized combination of mature elements of the myocardium, such as smooth muscle, blood vessels, fibroblasts, mature fat, and nerves, and are associated with a risk of SCD because of refractory ventricular tachycardia [12,74,75].

CMH is extremely rare with few cases reported in the literature, and etiopathogenesis is unclear but is believed to arise from a disorder in tissue development involving multiple myocardial cell lines [74].

The clinical presentation ranges from asymptomatic to symptoms of ventricular arrhythmias with incessant and refractory ventricular tachycardia, cardiac failure up to sudden cardiac death.

The diagnosis of CMH is difficult due to its rarity and nonspecific presentation.

Echocardiography reveals hyperechoic intracavitary masses with a predilection for the left ventricle [76].

CMH detects signal changes ranging from slightly hypointense to hyperintense on T2-weighted images, with early and delayed post-contrast enhancement [77].

The biopsy may be inconclusive due to the similarity of the hamartoma tissue to the surrounding myocardium.

Macroscopically, CMH appears as a well-demarcated pink-brown mass, predominantly located in the ventricular myocardium [12,78].

Histologically, CMH is characterized by a disorganized but well-circumscribed proliferation of cardiomyocytes, smooth muscle, blood vessels, fibroblasts, mature fat, and nerves [12,78].

Immunohistochemistry confirms the presence of all six histological components [78].

Surgical resection with epicardial mapping is successful in pediatric and adult patients. Catheter ablation is useful for treating relapses of ventricular tachycardia, and implantable defibrillators reduce the risk of SCD [74,75].

### 3.11. Hemangioma (NOS)

Cardiac hemangioma (CH) is a rare benign tumor of vascular origin, characterized by a proliferation of endothelial cells which leads to an increase in vascularization [79].

Based on its location and size, CH can cause severe complications such as syncope, embolism, and even sudden cardiac death from mechanical obstructions or arrhythmic disorders [79].

Etiopathogenesis is not completely known, but includes genetic factors, trauma, infections, and inflammatory processes [79].

CH represents 2–3% of cardiac tumors [80], manifests itself mainly in the fifth decade of life, but can appear at any age, and seems to affect females more [81].

CHs are divided into cavernous, capillary, and arteriovenous, and the cavernous hemangioma is the most frequent, followed by the capillary and the arteriovenous [82].

Symptoms vary from asymptomatic to dyspnea, angina, syncope, congestive heart failure, arrhythmias, and SCD. Clinical presentation depends on location, size, and growth [83].

Preoperative diagnosis is difficult. Echocardiography is the first-line test for detecting intracardiac masses. CT and MRI define the size, location, and relationship with adjacent structures. Coronary angiography is useful for evaluating vascularity [84].

On macroscopic examination, the CH is usually well circumscribed, soft in consistency, and has a smooth surface; the size is variable up to dimensions greater than 5 cm (Figure 6) [12,83].

Histologically, CH is characterized by vascular channels lined with benign endothelial cells [2,12]. Cavernous hemangioma is characterized by dilated vascular spaces with thin walls. Capillary hemangioma is characterized by small, regularly organized vessels. Arteriovenous hemangioma is characterized by malformations of arteries and veins [79,82,85]. Histotypes do not affect therapy [81].

On immunohistochemistry, CH is positive for vascular markers such as CD31, CD34, and, sometimes, ERG [12].

The treatment of choice is complete surgical resection, which is considered curative in most cases [86]; however, spontaneous regression is possible [87]. Medical therapy, with beta-blockers or corticosteroids, can be used in select unresectable cases. Radiation treatment is reserved for rare situations [88].

### 3.12. Conduction System Hamartoma

Conduction system hamartoma (CSH) is a rare benign hamartomatous lesion originating from cardiac Purkinje and Purkinje-like cells [89], characterized predominantly by benign tumors involving the cardiac conduction system, and is often associated with SCD [90,91,92,93].

CSH, formerly known as histiocytoid cardiomyopathy, has been changed to highlight its predominantly tumoral rather than cardiomyopathic nature [12,89].

It has a significant incidence in children under 2 years of age and mainly affects females [90,93].

It is associated with genetic mutations in the NDUF family genes involved in mitochondrial oxidative phosphorylation [12,94].

Clinically, it manifests itself with cardiomegaly, dilated or hypertrophic cardiomyopathy, life-threatening refractory ventricular arrhythmias, up to SCD [90,91,92,93].

Diagnosis is based on clinical signs, imaging, and histological confirmation. Macroscopically, CSH is characterized by tan-yellow subendocardial nodules; histologically it is characterized by polygonal cells with granular sarcoplasm and enlarged mitochondria, and has irregular reactivity for S100 on immunohistochemistry [12].

Treatment includes anti-arrhythmics, ablation, and mechanical support [95]. Recently, the use of carvedilol at high doses has shown efficacy in controlling refractory arrhythmias due to its action on ryanodine receptors [90].

### 3.13. Cystic Tumor of Atrioventricular Node

Cystic tumor of the atrioventricular node (CTAVN) is a rare congenital benign tumor, located in the region of the AV node, at the base of the interatrial septum, in Koch’s triangle, and is the most frequent cause of SCD associated with cardiac tumors [96,97,98].

Etiopathogenesis is linked to embryonic residues of endodermal development with differentiation towards a higher foregut phenotype [99].

CTAVN accounts for 2.7% of primary cardiac tumors and most commonly affects young women [97].

It clinically causes symptoms such as fatigue, dyspnea, syncope [97], and complete heart block, present in 65% of cases, or partial in 15% [100,101].

Macroscopically, CTAVN appears as a 2 mm to 2 cm multicystic mass, often not obvious, located between the coronary sinus ostium and the tricuspid septal leaflet [12,101].

Histologically, it is characterized by solid cysts and nests of nonciliated, cuboidal or squamous epithelial cells, immersed in a fibrous stroma with collagen and elastin, without atypia or signs of malignancy [12,101].

Immunohistochemistry is positive for epithelial markers such as CK5/6, CK7, EMA, CEA, and neuroendocrine markers such as chromogranin, synaptophysin, TTF1, with low Ki-67 proliferative index < 2% [101].

The diagnosis is based on high-resolution cardiac CMR or CT imaging and on autopsy analysis in cases of sudden death, with sections of the conduction system [102].

The recommended therapy is surgical excision, and in some cases an implantable defibrillator is considered to reduce the risk of SCD post-surgery [101].

## 4. Malignant Tumors

### 4.1. Angiosarcoma

Angiosarcoma is a rare primary cardiac malignant tumor. It is the most frequent cardiac sarcoma and causes SCD due to hemopericardium, cardiac tamponade, or malignant arrhythmias [2,3,12].

Angiosarcoma predominantly affects males between 40 and 50 years of age with an extremely low incidence, but represents 30% of malignant cardiac sarcomas [103], and is in 75% of the right atrium [104], with frequent metastases to the lungs, bones, liver, and brain [105].

Angiosarcoma arises from the vascular endothelium and is characterized by extensive vasculature and pleomorphic malignant endothelial cells [103].

The major molecular alterations in cardiac angiosarcoma are mutations in genes such as KDR, which encodes a VEGF receptor tyrosine kinase, and PLCG1, involved in phosphoinositide signaling, both associated with resistance to VEGF/KDR targeted therapies [106,107]. However, less frequent than in soft tissue, angiosarcomas are MYC amplifications and TP53 mutations, and in a minority of cases anomalies in the RAS pathway are observed (KRAS, H/K/N-RAS) [103]. Furthermore, mutations in the POT1 gene responsible for telomere protection are detected in both familial and sporadic cardiac angiosarcomas, suggesting a possible link to Li-Fraumeni-type syndromes [108].

Clinically it presents with dyspnea, chest pain, palpitations, right heart failure, pericardial effusion, cardiac tamponade, valvular dysfunction, and sudden cardiac death [2,103].

The diagnosis is carried out by transthoracic and transesophageal echocardiography, completed by cardiac magnetic resonance imaging and computed tomography to evaluate the extension and nature of the tumor, and PET/CT angiography useful for characterizing metabolic activity and differentiating malignant from benign tumors.

Macroscopically (Table 2), angiosarcoma appears as a hemorrhagic, infiltrative mass, often with necrosis [12].

Microscopically (Table 2), cardiac angiosarcoma is characterized by anastomotic vascular channels formed by malignant cells, solid areas of spindle cells, and regions of anaplastic cells, and foci of endothelial tufts, while calcification is absent [12,109].

Immunohistochemistry (Table 2) is positive for endothelial markers such as CD31, CD34, factor VIII-related protein, von Willebrand factor, cytokeratin, vimentin, BNH9, p53, Ki67, alpha-smooth muscle actin, and Wilms tumor 1 [12,103].

Complete surgical resection is the best chance for survival but is often not feasible due to local extension and metastasis. Neoadjuvant therapy with chemotherapy and radiation therapy may improve operability. Palliative treatment includes systemic chemotherapy and, in select cases, radiotherapy. Despite interventions, the prognosis remains poor, with a median survival of around 6 months [110,111].

### 4.2. Leiomyosarcoma (NOS)

Cardiac leiomyosarcoma is a rare malignant mesenchymal neoplasm; it is highly aggressive and is associated with sudden cardiac death due to arrhythmias, heart failure, or cardiac tamponade [112].

It represents less than 0.25% of all cardiac tumors [113], predominantly affecting females with an average age at diagnosis of around 45 years; in 50% of cases, it is in the left atrium [112].

The symptoms vary according to size, location, and stage, with dyspnea, chest pain, cough, right heart failure, valvular stenosis, and cardiac tamponade [114,115].

Transthoracic echocardiography is the gold standard for identifying cardiac masses and evaluating size, position, and mobility, while transesophageal echocardiography and cardiac magnetic resonance imaging are useful for defining anatomical details, infiltration, and vascularization [116].

Macroscopically, the tumor presents as a sessile, often mucoid mass, while histologically it is characterized by compact bundles of spindle-shaped cells with blunt nuclei, necrotic regions and mitotic figures, and immunohistochemistry, it is positive for desmin and smooth muscle actin (Table 2 and Figure 7) [12].

Treatment is surgical, with palliative resection which alleviates symptoms and improves survival. Adjuvant radiation therapy and chemotherapy can be considered, but their role remains controversial. The prognosis is unfavorable, with an average survival of approximately 6 months if untreated, and increases up to 24–34 months with complete resection [115,117,118].

### 4.3. Pleomorphic Sarcoma

Undifferentiated pleomorphic sarcoma (UPS) is a high-grade malignant tumor originating from mesenchymal cells that is very aggressive and associated with episodes of sudden cardiac death, especially when involving the left atrium [2,119,120].

UPS, previously classified as malignant fibrous histiocytoma [120], represents 20% of soft tissue sarcomas and has a global incidence estimated between 0.08 and 1 per 100,000 inhabitants [121].

UPS primarily affects the left atrium, without gender predilection, between the fourth and sixth decade [12], with dyspnea, palpitation, heart failure, or constitutional symptoms and complications such as arrhythmia, embolic events, pericardial effusion, or distant metastases [122].

Clinically and on imaging, UPS can mimic myxoma, but unlike myxoma, it can form multiple masses, extend into the pulmonary veins, and impinge on the mitral valve [12].

Macroscopically (Table 2), UPS may appear as an infiltrative or necrotic mass with areas of hemorrhage. Histologically (Table 2), it is characterized by epithelioid or spindle-shaped cells (Figure 8), but the exclusion of leiomyosarcoma or synovial sarcoma is important; furthermore, it can also be characterized by a polypoid tumor with absent or minimal myxoid or fibrotic matrix in the left atrium [11,12,122].

The diagnosis is one of exclusion, as UPS has no specific tissue or immunohistochemical characteristics. The classification follows the FNCLCC system, based on tumor differentiation, mitosis, and necrosis [12].

The treatment of choice is complete surgical resection with adequate margins, often associated with adjuvant radiotherapy for large, high-grade tumors > 5 cm. Chemotherapy, with agents like doxorubicin and ifosfamide, is reserved for metastatic or inoperable cases. Newer approaches include immunotherapy with checkpoint inhibitors such as pembrolizumab and targeted therapies, such as larotrectinib for tumors with NTRK gene fusions [123,124,125,126].

The prognosis of UPS is generally poor, with a median survival rate of less than one year in advanced cases [123], underscoring the importance of early diagnosis and a multidisciplinary approach to treatment.

### 4.4. Metastatic Neoplasm

Cardiac metastases are much more frequent than primary cardiac tumors, but they are often diagnosed postmortem as asymptomatic. An autopsy study has highlighted that approximately 4–10% of patients suffering from solid tumors present metastases to the heart [3] and cause sudden cardiac death [126,127].

Primary tumors of the lung, pleural mesothelioma, skin melanoma, breast, esophagus, and hematological neoplasms frequently metastasize to the heart; less frequently, they metastasize to the uterine, vulva, thyroid, liver, colorectal, renal, and head and neck cancer [2,3,7,12,128].

They involve the pericardium in two-thirds of cases, followed by the myocardium in a third of cases and, less frequently, the endocardium [3].

The spread of neoplastic cells occurs through various mechanisms, such as direct extension, hematogenous dissemination, lymphatic propagation and, in selected cases, transvenous spread [2].

Clinically, cardiac metastases manifest themselves with symptoms that vary according to location and extent, such as dyspnea, chest pain, hypotension disproportionate to radiological findings, and signs of cardiac tamponade in the case of malignant pericardial effusion and systemic symptoms, such as fatigue and weight loss, or embolic complications if it affects the endocardium.

The diagnosis of cardiac metastases can be challenging. Transthoracic echocardiography and cardiac magnetic resonance imaging are useful tools for detecting intracardiac masses, while pericardial fluid cytology, obtained via pericardiocentesis, can reveal the presence of tumor cells, but has limited sensitivity and may result in false negatives in 15% of cases [129].

Treatment is often palliative, aiming to relieve symptoms and improve the patient’s quality of life. In selected cases, surgery for mass removal or management of pericardial effusion is required. However, the general prognosis of cardiac metastases remains poor, reflecting the advanced nature of the primary disease [130,131].

Table 2 describes the macroscopic and histological features.

### 4.5. Hematolymphoid Tumors

Primary cardiac lymphomas are rare, accounting for <2% of cardiac tumors, but are potentially lethal and can cause sudden cardiac death [132].

The most frequent histological type is diffuse large B-cell lymphoma (DLBCL), while few cases are described in the literature of fibrin-associated DLBCL (FA-DLBCL) which is a type of DLBCL associated with chronic inflammation [132].

Etiopathogenesis includes the abnormal proliferation of lymphocytes often in association with viral infections such as EBV or immunosuppressive states, as in patients with HIV/AIDS or subjected to immunosuppressive therapies.

Clinically, they manifest with nonspecific symptoms such as dyspnea and chest pain, often accompanied by pericardial effusion, tamponade, heart failure, and atrioventricular block [133,134,135,136].

The diagnostic process is based on multimodal imaging with echocardiography, CMR, PET, and biopsy confirmation [133,135].

Macroscopically (Table 2), cardiac lymphomas appear as solid masses, with a friable consistency, with an infiltrative pattern frequently involving the myocardium, pericardium, or both, which can cause local deformities and significant enlargements of the cardiac tissue [2,12].

Histologically (Table 2), cardiac lymphomas are characterized by atypical lymphoid cells, typically large in DLBCL, with vesicular nuclei, prominent nucleoli, and variable cytoplasm, frequently associated with mitosis, areas of necrosis, and infiltration of surrounding tissues, with destruction of myocardial architecture [2,12].

At immunohistochemistry (Table 2), it is positive for B markers such as CD20 and CD79a; the Ki-67 proliferative index is typically elevated due to aggressive tumor growth; markers such as BCL-2 and BCL-6 may be present in some subtypes, while positivity for EBER is indicative of an association with Epstein–Barr virus [2,12].

Therapy consists of chemotherapy regimens such as the CHOP protocol combined with rituximab, and radiotherapy in selected cases, but the overall prognosis remains severe due to cardiovascular complications and tumor progression [137,138,139].

## 5. Physiopathological Mechanisms of Sudden Cardiac Death in Cardiac Tumors

SCD in patients with CTs represents a complex clinical condition, resulting from multiple pathophysiological mechanisms. In fact, both primary and metastatic CTs can compromise cardiac activity through electrical alterations, embolization, mechanical compression, and vascular compromise (Table 3).

Arrhythmogenesis is a key mechanism by which tumors infiltrating the myocardium or cardiac conduction system can induce lethal arrhythmias, resulting in severe bradyarrhythmias, conduction blocks, ventricular tachyarrhythmias, or ventricular fibrillation. The presence of a tumor mass can directly interfere with the normal electrical conduction circuits of the heart, altering the transmission of electrical impulses and promoting the development of unstable cardiac rhythms. Furthermore, myocardial fibrosis is secondary to tumor growth and the cardiotoxic effects of oncology therapies further contribute to the electrical instability of the heart. Certain chemotherapy drugs, such as anthracyclines, can induce cumulative myocardial damage, increasing the risk of heart failure and fatal arrhythmias. Thoracic radiotherapy can also cause fibrosis of the cardiac tissue, with consequent alterations in electrical conduction and reduction in myocardial contractility.

Another notable mechanism is tumor embolization. Some CTs, such as atrial myxomas, can release fragments into the systemic circulation, resulting in arterial embolization and vascular occlusion. At the cerebral level, this process can cause ischemic stroke with serious neurological consequences, while embolization in the coronary arteries can lead to acute myocardial infarctions, with subsequent development of heart failure and increased risk of SCD. Furthermore, the endothelial inflammation induced by the presence of the tumor can favor the formation of thrombi, exacerbating the thromboembolic risk and worsening the patient’s prognosis.

Vascularized tumors, such as cardiac angiosarcomas, can cause intrapericardial hemorrhages resulting in cardiac tamponade. This condition occurs when blood rapidly accumulates in the pericardial sac, compressing the heart and preventing its normal diastolic filling. Cardiac tamponade represents a medical emergency and, if not treated promptly with pericardial drainage or surgery, can rapidly lead to hemodynamic collapse and death.

An additional risk factor is represented by the mechanical compression exerted by the CTs on the cardiac structures. Intracavitary tumor masses can reduce the volume available for ventricular filling, hindering normal ejection function and causing congestive heart failure or cardiogenic shock. If the tumor is located near the heart valves, it can compromise blood flow and lead to valve insufficiency. In some cases, tumors infiltrating the myocardial wall can cause structural weakening of the heart, predisposing to cardiac rupture, a catastrophic event that rapidly leads to sudden death.

Advanced imaging techniques such as transesophageal echocardiography and cardiac magnetic resonance imaging are critical to identifying tumors at risk of causing fatal events.

The use of devices such as implantable defibrillators may be indicated for patients with tumors that compromise the conduction system. Additionally, regular monitoring of patients with already diagnosed cancers is crucial to prevent sudden complications.

The management of CTs associated with SCD requires a multidisciplinary approach with surgical, oncological, and symptomatic treatment.

The treatment of choice for many benign tumors is complete surgical removal, when possible, which significantly reduces the risk of SCD.

Instead, the combination of surgery, chemotherapy, and radiotherapy in malignant tumors can improve the overall outcome, but the risk of SCD often remains high.

Furthermore, symptomatic therapies to control arrhythmias, manage heart failure, and prevent embolic complications are fundamental to reducing mortality.

The development of new therapies, such as molecular inhibitors and immunotherapy, could offer new opportunities to improve survival and reduce complications, and research on specific biomarkers could help identify patients at higher risk.

The integration of clinical, diagnostic, and therapeutic approaches in a multidisciplinary context is essential for reducing the risk of sudden events and improving the quality of life.

## 6. Emerging Therapies and Patient Selection for Surgical and Non-Surgical Management

In recent years, the introduction of emerging therapies, such as immunotherapy and targeted molecular treatments, has revolutionized the management of TCs, particularly malignant neoplasms such as angiosarcoma and leiomyosarcoma. Immunotherapy, based on immune checkpoint inhibitors, has shown promise in improving survival in certain sarcoma subtypes [140]. These drugs work by blocking the mechanisms by which tumor cells evade the immune system, allowing a more effective immune response against neoplastic cells. However, their application in TCs is still under investigation, given the rarity of these neoplasms and the need for more robust clinical data.

In parallel, molecularly targeted treatments are emerging as therapeutic options for CTs with specific genetic alterations. For example, mutations in the *KDR* gene, which encodes a VEGF receptor, have been identified in some angiosarcomas [141]. Furthermore, the discovery of mutations in the POT1 gene in cardiac sarcomas suggests a potential role for telomere-targeted therapies in the management of these neoplasms [142]. However, because these treatments can have significant side effects, such as cardiovascular toxicity and endothelial dysfunction, their use requires careful monitoring by a multidisciplinary team.

A crucial aspect in the management of CTs is the selection of patients for surgical versus non-surgical management. The decision to proceed with surgery depends on several factors, including the size, location, and histotype of the tumor, as well as the clinical status of the patient. In cases of benign tumors, such as myxomas, surgery represents the first line of treatment, as complete resection leads to an excellent prognosis [143]. However, in malignancies, such as angiosarcomas, surgery is often reserved for cases in which a complete resection with negative margins can be achieved, reducing the risk of recurrence. In these patients, the combination of surgery and adjuvant therapies, such as chemotherapy and radiotherapy, is often required to improve survival.

For inoperable patients, management focuses on systemic treatments and palliative support. In cases where surgical resection is associated with a high risk of complications or limited benefit, non-surgical approaches, such as stereotactic radiotherapy or embolization of tumor arteries, are opted for to control neoplastic growth and reduce symptoms.

Furthermore, new diagnostic modalities such as PET/CT and advanced molecular imaging are taking on an increasingly central role in the differentiation between benign and malignant lesions and in the evaluation of tumor activity [144,145]. These techniques allow a more precise characterization of the tumor, supporting therapeutic decisions and improving patient selection for surgery or systemic therapies.

The decision between surgery and systemic therapy must be personalized and based on a multidisciplinary assessment involving cardiologists, oncologists, and cardiac surgeons, to maximize the therapeutic benefit and improve the patient’s quality of life.

## 7. Importance of the Multidisciplinary Team

The integration between cardiologists, oncologists, and surgeons is essential to improve the clinical outcomes of patients suffering from CTs, as it allows a global approach to diagnosis, treatment, and follow-up [146].

For example, in patients with cardiac sarcomas, early diagnosis using advanced imaging, such as CMRI and PET, requires close collaboration between cardiologists and oncologists to characterize the extent of the tumor and plan treatment.

Cardiovascular surgeons, in turn, play an essential role in tumor resection, often necessary to prevent blockage of the heart chambers or valves. However, given the aggressive nature of many cardiac neoplasms, surgery alone is not sufficient and must be followed by adjuvant therapies, such as chemotherapy and radiotherapy, the impact of which on the cardiovascular system must be monitored by cardiologists to prevent cardiac toxicity. A concrete example is the treatment of patients with cardiac angiosarcomas, in which a multidisciplinary team can determine the feasibility of surgical resection and, in inoperable cases, optimize systemic therapy to slow disease progression without compromising cardiac function.

Even in benign tumors, such as atrial myxoma, collaboration between specialists is crucial to avoid complications. Cardiologists intervene in the diagnosis using echocardiography, surgeons remove the mass, and oncologists evaluate the possible presence of associated genetic syndromes, such as Carney syndrome, which require long-term monitoring.

Finally, in cancer patients undergoing cardiotoxic chemotherapy, the role of the cardiologist is fundamental to prevent and manage heart failure or arrhythmia, while the surgeon can intervene in selected cases with implants of cardiac assistance devices.

These examples demonstrate how a multidisciplinary approach not only improves survival, but also the quality of life of patients, reducing the risk of complications and optimizing treatment strategies tailored to each clinical case.

## 8. Future Research Directions

An emerging area of research concerns the development of personalized therapies based on specific genetic profiles of CTs. The identification of new biomarkers could enable more effective and less toxic treatments, improving patient outcomes. Furthermore, the use of nanotechnology for targeted drug delivery could revolutionize the therapeutic management of these tumors. Collaboration between biomedical engineers, oncologists, and cardiologists will be key to bringing these innovations from research to clinical practice.

Innovative technologies such as nanocarriers are being developed for the selective release of drugs into tumor tissues, reducing systemic toxicity and increasing therapeutic efficacy. Furthermore, the combined use of advanced molecular imaging and AI-based predictive models could enable more timely and accurate diagnoses, improving risk stratification in patients with CTs.

Another area of great interest is the study of the interactions between the cardiac tumor microenvironment and the immune system, which could lead to the development of new therapeutic strategies based on the modulation of the immune response. Research into gene therapies, such as genome editing using CRISPR, could open up new perspectives for treating tumors with specific mutations.

Finally, the creation of international registries and the implementation of multicenter clinical trials dedicated to CTs could provide more robust data and facilitate the development of evidence-based treatment guidelines.

## 9. Conclusions

CTs are rare tumors that represent a significant challenge for both diagnosis and treatment, with important consequences on the prognosis of patients. Effective management of these malignancies requires a multidisciplinary approach to reduce the risk of SCD.

Future research should focus on improving personalized therapeutic options, taking advantage of technological innovation and a deeper understanding of the biological mechanisms underlying these pathologies, and with an integrated approach and multidisciplinary collaboration, it is possible to significantly improve the outlook for CT patients.

## Figures and Tables

**Figure 1 cancers-17-00669-f001:**
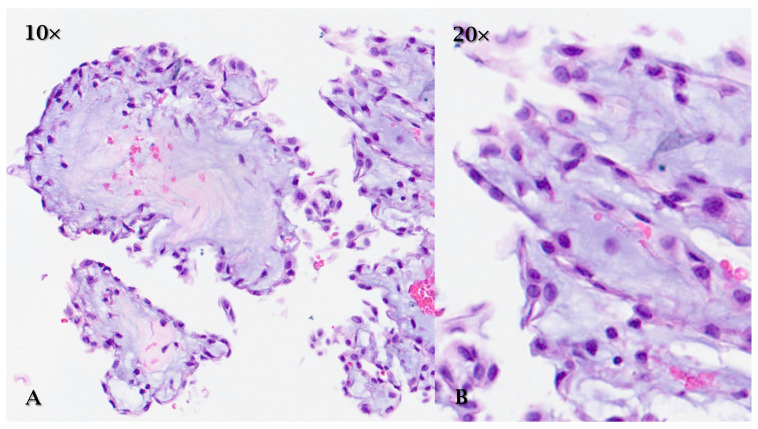
HE (**A**) (10×) and (**B**) (20×): papillary fibroelastoma is characterized by fronds covered by a single layer of endothelial cells that cover a central core of avascular fibroelastic connective tissue. The core is composed of an extracellular matrix rich in collagen, elastin, and proteoglycans.

**Figure 2 cancers-17-00669-f002:**
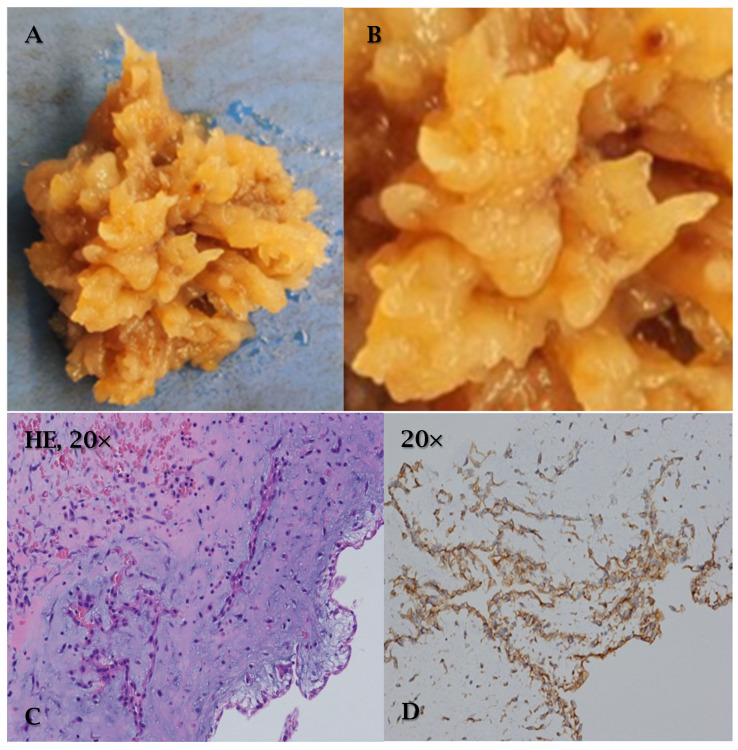
Cardiac myxoma. A 56-year-old patient with episodes of syncope and transient ischemic stroke. Transesophageal echocardiography revealed a left atrial myxoma with risk of embolization. After removal surgery, the patient remained asymptomatic with negative 5-year follow-up. (**A**,**B**) Myxoma appears as a villiform architecture and friable mass. (**C**) Myxoma is characterized by isolated or grouped cellularity, immersed in a myxoid matrix. (**D**) Positive immunohistochemistry for CD34.

**Figure 3 cancers-17-00669-f003:**
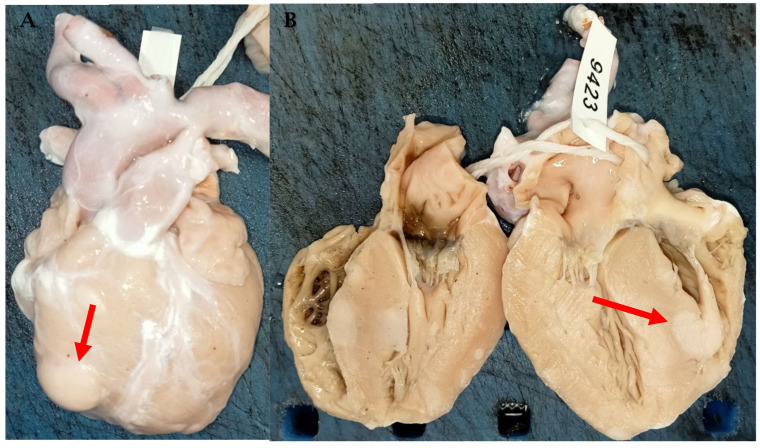
Cardiac rhabdomyoma. An 8-month abortion with genetic diagnosis of tuberous sclerosis. (**A**,**B**): Rhabdomyoma appears as multiple well-circumscribed nodules (red arrow), which develop primarily in the myocardium.

**Figure 4 cancers-17-00669-f004:**
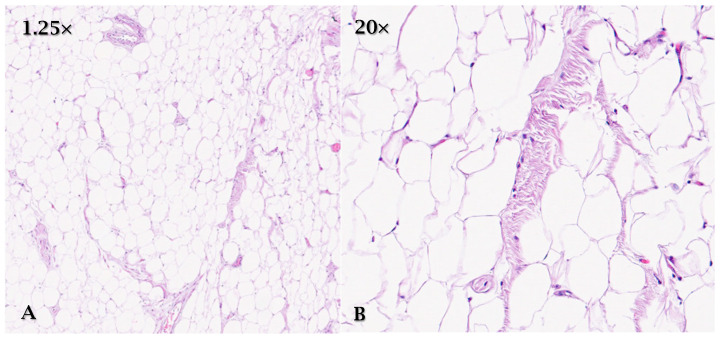
HE, (**A**) (1.25×) and (**B**) (20×): cardiac lipoma is characterized by mature adipocytes free of atypia.

**Figure 5 cancers-17-00669-f005:**
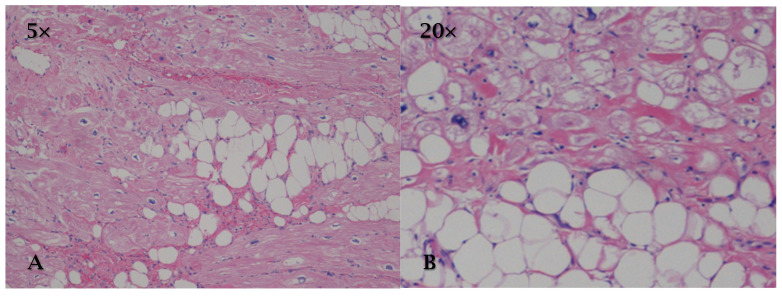
HE, (**A**) (5×) and (**B**) (20×): lipomatous hypertrophy of the atrial septum consists of a mixture of adipose tissue including interspersed by fetal fat cells, fibrous tissue, and hypertrophied myocytes.

**Figure 6 cancers-17-00669-f006:**
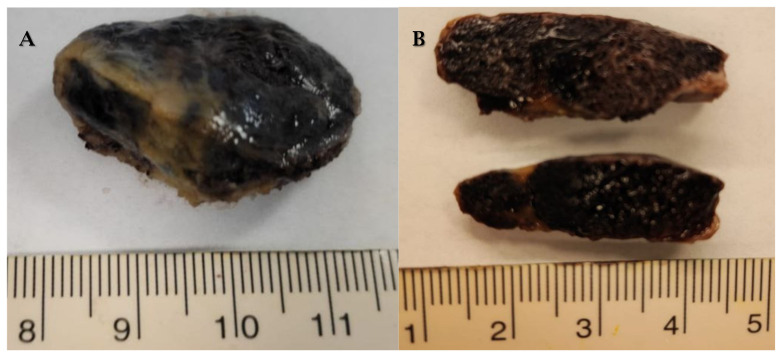
Cardiac hemangioma is well circumscribed, soft in consistency, and a smooth surface (**A**), and is brownish in color when cut (**B**).

**Figure 7 cancers-17-00669-f007:**
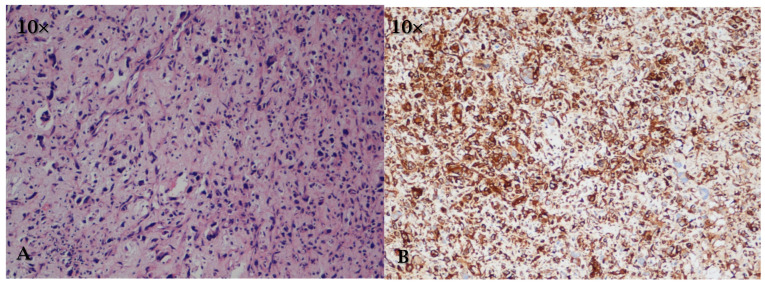
(**A**) (HE, 10×): cardiac leiomyosarcoma is histologically characterized by compact bundles of spindle-shaped cells with blunt nuclei, necrotic regions, and mitotic figures. (**B**) (IHC, 10×): cardiac leiomyosarcoma is positive for smooth muscle actin.

**Figure 8 cancers-17-00669-f008:**
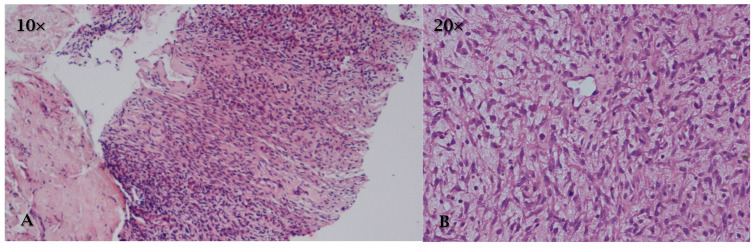
HE, (**A**) (10×) and (**B**) (20×): undifferentiated pleomorphic sarcoma is characterized by epithelioid or spindle-shaped cells.

**Table 1 cancers-17-00669-t001:** Classification cardiac tumors.

Tumor	Subtype
**Benign tumors**	
	Papillary fibroelastoma
	Myxoma (NOS)
	Fibroma (NOS)
	Rhabdomyoma (NOS)
	Adult cellular rhabdomyoma
	Lipoma (NOS)
	Lipomatous hypertrophy of the atrial septum
	Lipomatous hamartoma of atrioventricular valve
	Hamartoma of mature cardiac myocytes
	Mesenchymal cardiac hamartoma
	Hemangioma (NOS) - Venous hemangioma - Capillary hemangioma - Arteriovenous hemangioma - Cavernous hemangioma
	Conduction system hamartoma
	Cystic tumor of atrioventricular node
**Malignant tumors**	
	Angiosarcoma
	Leiomyosarcoma (NOS)
	Pleomorphic sarcoma
	Metastatic neoplasm
	Hematolymphoid tumors: - Diffuse large B-cell lymphoma (NOS) - Fibrin-associated diffuse large B-cell lymphoma

**Table 2 cancers-17-00669-t002:** Macroscopic and histological features of malignant cardiac tumors.

Malignant Cardiac Tumor	Macroscopic Characteristics	Histological Features
**Angiosarcoma**	Infiltrating, hemorrhagic mass, often with necrosis, is localized mainly in the right atrium.	Anastomotic vascular channels consist of malignant cells, solid areas of spindle cells, and foci of anaplastic cells. Positive for endothelial markers such as CD31, CD34, factor VIII related protein, vimentin, p53, Ki67, and smooth muscle actin.
**Leiomyosarcoma (NOS)**	Sessile mass, often mucoid, is usually located in the left atrium.	Compact fascicles of spindle-shaped cells with blunt nuclei, necrotic areas, and mitotic figures. Positive for desmin and smooth muscle actin.
**Pleomorphic** **sarcoma** **(UPS)**	Infiltrating or necrotic mass with areas of hemorrhage, typically localized in the left atrium.	Epithelioid or spindle-shaped cells. Exclusion of sarcoma leiomas or synovial sarcoma. The diagnosis is one of exclusions, with FNCLCC classification system based on differentiation, mitosis and necrosis.
**Metastatic** **neoplasm**	Solid, often friable, infiltrative mass involving the pericardium, myocardium, or endocardium. It frequently involves pericardium, with metastases from the lung, melanoma, breast, esophagus, and others.	Atypical neoplastic cells, often with prominent nuclei and mitosis. Necrosis and infiltration of the surrounding tissue. Diagnosis confirmed through biopsy, with positive immunohistochemistry for specific markers depending on the primary tumor.
**Hematolymphoid tumors**	Massive solid masses, friable consistency, often infiltrative, affecting the myocardium and pericardium. They can cause local deformities and significant enlargement of cardiac tissues. Frequent pericardial and myocardial involvement.	Atypical lymphocyte cells, large in the case of LDLBCL, with vesicular nuclei and prominent nucleoli. Positivity for B markers such as CD20, CD79a, with elevated Ki-67 proliferative index. Also positive were BCL-2, BCL-6 and EBER. Necrosis and infiltration of surrounding tissues are common, with destruction of myocardial architecture.

**Table 3 cancers-17-00669-t003:** Pathophysiological mechanisms of SCD.

Mechanism	Description	Consequences
**Electrical alterations and arrhythmias**	Infiltration of the myocardium or cardiac conduction system, myocardial fibrosis due to tumor, or oncological therapies (chemotherapy/radiotherapy).	Severe bradycardia, conduction blocks, ventricular tachyarrhythmias, ventricular fibrillation, and SCD.
**Tumoral** **embolization**	Detachment of tumor fragments (e.g., atrial myxomas) with arterial embolism.	Ischemic stroke, myocardial infarction, heart failure, and SCD.
**Vascular compromise and** **cardiac tamponade**	Vascularized tumors (e.g., angiosarcomas) can cause pericardial bleeding.	Cardiac compression reduced diastolic filling, hemodynamic collapse, and SCD.
**Mechanical** **compression**	Intracardiac tumor masses or heart valve involvement.	Reduced ventricular filling, congestive heart failure, cardiogenic shock, valvular insufficiency, risk of cardiac rupture, and SCD.

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
