# Peer review of "Cardiac Tumors Causing Sudden Cardiac Death: A State-of-the-Art Review in Pathology"

_cancers, 2025, doi:10.3390/cancers17040669_

Round 1
Reviewer 1 Report
Comments and Suggestions for Authors
This review article, "Cardiac Tumors Causing Sudden Cardiac Death: A State-of-the-Art Review in Pathology," offers an extensive and thorough examination of the various types of cardiac tumors, their pathophysiology, and their significant role in sudden cardiac death (SCD). The authors have successfully compiled and analyzed a wide array of information, making this a valuable resource for both clinicians and researchers.
Strengths:
1. Comprehensive Coverage: The article provides detailed insights into the classification, diagnosis, treatment, and potential complications of both benign and malignant cardiac tumors. This comprehensive approach ensures that readers gain a deep understanding of the subject matter.
2. Diagnostic Techniques: The discussion on advanced imaging techniques and their roles in diagnosing cardiac tumors is particularly enlightening. It emphasizes the importance of precise and early diagnosis to improve patient outcomes.
3. Clinical Relevance: By focusing on the implications of cardiac tumors in the context of SCD, the review highlights the urgency and importance of timely intervention and management. This focus enhances the article's relevance to clinical practice.
Areas for Improvement:
1. Inclusion of Latest Guidelines: Incorporating the most recent clinical guidelines and recommendations from leading cardiology and oncology societies would add further credibility and practical value to the review.
2. Case Studies: Including illustrative case studies or real-world examples could provide a practical dimension to the theoretical information presented, helping readers relate to the clinical scenarios discussed.
3. Future Research Directions: A more explicit section on future research directions would be beneficial. Highlighting areas where further studies are needed could guide researchers and stimulate new investigations in the field.
4. Visual Aids: Adding more visual aids, such as diagrams, imaging examples, and histological slides, would enhance the reader's understanding and retention of the information.
Author Response
Dear Reviewer 1,
Thank you for your precious comments and suggestions. In view of them we have modified the paper in this way:
1) We added new references. Updated guidelines, as far as we know, are not still available because this area of medicine has limited trials and evidence to base decision making.
2) We added new images concerning clinical cases: figure 2 (including histology and immunohistochemistry), figure 3 (description of the case and related gross anatomy), figure 6 (gross anatomy).
3) A new paragraph (8) concerning future research direction has been added to the text and evidenced in red.
4) Two new tables (2 and 3) together with the above cited images have been added to the text to enhance the reader's understanding as suggested.
Kind regards
Reviewer 2 Report
Comments and Suggestions for Authors
This manuscript provides a comprehensive and well-structured review of cardiac tumors (CTs), particularly their association with sudden cardiac death (SCD). It is an important contribution to the field, addressing both the diagnostic and therapeutic challenges associated with these rare but critical conditions. While the manuscript is of high quality, there are areas that require clarification and elaboration to enhance its clinical relevance.
1. While the manuscript discusses the clinical and histological aspects of cardiac tumors, the pathophysiological mechanisms leading to sudden cardiac death (e.g., arrhythmogenesis, embolization) could be more thoroughly explored. This would provide readers with a deeper understanding of how these tumors result in fatal outcomes.
2. Although echocardiography, MRI, and CT are well-discussed, the role of newer diagnostic modalities such as PET/CT and advanced molecular imaging should be incorporated. These techniques are increasingly used to differentiate between benign and malignant lesions and to assess tumor activity.
3. The section on treatment could benefit from a more detailed discussion of emerging therapies, such as immunotherapy and targeted molecular treatments. Additionally, considerations regarding patient selection for surgical versus non-surgical management could provide more actionable insights for clinicians.
4. While the manuscript emphasizes the importance of a multidisciplinary team, specific examples of how collaboration between cardiologists, oncologists, and surgeons impacts patient outcomes would strengthen the argument.
Author Response
Dear Reviewer 2,
Thank you for your appreciated comments. We have modified the text as follows:
1) We provided new insights into pathogenetic mechanisms leading to sudden death in cardiac tumours: paragraph 5 evidenced in yellow.
2) We added few lines concerning the use of PET/TC and molecular imaging as reported in paragraph 6 evidenced in red and related references n. 150 and 151.
3) Emerging therapies and considerations about patient selection have been included in paragraph 6 and evidenced in light blue (references from 146 to 149).
4) We added a paragraph to underline the importance of a multidisciplinary team for the management of cardiac tumours enriched with examples: paragraph 7 evidenced in green and reference 152.
Kind regards